# Effect of Skin Ion Channel TRPM8 Activation by Cold and Menthol on Thermoregulation and the Expression of Genes of Thermosensitive TRP Ion Channels in the Hypothalamus of Hypertensive Rats

**DOI:** 10.3390/ijms23116088

**Published:** 2022-05-29

**Authors:** Irina P. Voronova, Galina M. Khramova, Anna A. Evtushenko, Tamara V. Kozyreva

**Affiliations:** 1Department of Thermophysiology, Scientific Research Institute of Neurosciences and Medicine, 630117 Novosibirsk, Russia; galina.khramova2015@yandex.ru (G.M.K.); a.a.evtushenko@yandex.ru (A.A.E.); kozyreva@physiol.ru (T.V.K.); 2Department of Physiology, Novosibirsk State University, 630090 Novosibirsk, Russia

**Keywords:** cold/menthol activation of TRPM8, gene expression, thermoregulation, hypertension

## Abstract

ISIAH (inherited stress-induced arterial hypertension) rats are characterized by high blood pressure and decreased *Trpm8* gene expression in the anterior hypothalamus. Thermosensitive ion channel TRPM8 plays a critical role in the transduction of moderately cold stimuli that give rise to cool sensations. In normotensive animals, the activation of skin TRPM8 is known to induce changes in gene expression in the hypothalamus and induce alterations of thermoregulatory responses. In this work, in hypertensive rats, we studied the effects of activation of the peripheral TRPM8 by cooling and by application of a 1% menthol suspension on (1) the maintenance of body temperature balance and (2) mRNA expression of thermosensitive TRP ion channels in the hypothalamus. In these hypertensive animals, (1) pharmacological activation of peripheral TRPM8 did not affect the thermoregulatory parameters either under thermoneutral conditions or during cold exposure; (2) the expression of *Trpm8* in the anterior hypothalamus approximately doubled (to the level of normotensive animals) under the influence of (a) slow cooling and (b) at pharmacological activation of the peripheral TRPM8 ion channel. The latter fact seems the quite important because it allows the proposal of a tool for correcting at least some parameters that distinguish a hypertensive state from the normotensive one.

## 1. Introduction

According to a number of authors at present [1,2,3,4,5], thermosensitive ion channels, including the TRP (transient receptor potential) family of channels, are primary detectors of temperature changes in warm-blooded animals and are claimed to be the molecular basis of thermal sensitivity. Moreover, the ability of these channels to be activated by temperature changes and by nonthermal stimuli (chemical agonists) of natural and artificial origin makes these channels promising targets for medical treatments [2,3,6,7,8].

Our previous studies on mRNA expression of thermosensitive TRP ion channels in the hypothalamus—the brain structure regulating visceral functions of the body—showed that normotensive and hypertensive animals differ in the level of *Trpm8* expression [9]. This gene encodes the TRPM8 thermosensitive channel that is permeable for calcium and sodium ions. This channel is activated by cold and menthol and is postulated to play a critical role in the transduction of moderate cold stimuli that give rise to cool sensations [1,5,10]. According to our previous results, the mRNA level of the *Trpm8* gene in the anterior hypothalamus in hypertensive rats is more than twofold lower than that in normotensive animals [9]. Later research has shown that, in hypertensive animals, the mRNA level of this gene is also low at the periphery: in the spleen (an organ of the immune system) [11].

Our studies in normotensive animals have revealed that changes in the expression of some genes of thermosensitive TRP ion channels in the hypothalamus can be caused by short-term cooling of a warm-blooded animal [12] as well as by menthol-induced activation of the peripheral skin TRPM8 [13]. In addition, it was found in that study that the activation of the peripheral TRPM8 by its agonist menthol leads to an increase in oxygen consumption and a decrease in the respiratory exchange ratio under thermoneutral conditions. These data indicate that nonthermal activation of TRPM8 even under thermoneutral conditions causes the same physiological responses in warm-blooded animals as does cold exposure: acceleration of overall metabolism and greater use of lipids as an oxidation substrate [14,15]. Pharmacological preactivation of the peripheral TRPM8 also influences the development of thermoregulatory responses of the organism to subsequent cooling, by decreasing the thresholds of all thermoregulatory reactions without changing the sequence of their initiation [16,17].

Hypertensive animals, as mentioned above, are characterized by reduced level of *Trpm8* mRNA in the anterior hypothalamus. In this regard, the question arises: does this feature of hypertensive animals affect their physiology, in particular, their thermoregulatory responses and mRNA expression of thermosensitive TRP ion channels in the hypothalamus in response to stimulation (temperature and pharmacological) of peripheral ion channel TRPM8?

Therefore, our aims were to find out how, during such a pathology as arterial hypertension (in a rat model), the activation of peripheral cold-sensitive ion channel TRPM8 by cooling (thermal activation) or by menthol (pharmacological activation) affects the following:(1)The parameters of cold-defense responses;(2)mRNA expression of thermosensitive TRP ion channels (TRPA1, TRPM8, TRPV1, TRPV2, TRPV3, and TRPV4) in the hypothalamus, the center of visceral function regulation.

## 2. Results

Hypertensive rats used in our study weighed 313.0 ± 4.2 g (mean ± SEM). Their blood pressure was 176.8 ± 0.8 mm Hg (mean ± SEM).

### 2.1. Parameters of Temperature Homeostasis

Under thermoneutral conditions, physiologically significant differences in the temperature homeostasis parameters were not found between the groups of hypertensive rats treated with either saline or menthol; at the beginning of cold exposure, the analyzed parameters of temperature homeostasis were not different between these groups (Table 1).

Under cooling. In the experimental groups, the cooling caused a set of reactions aimed at ensuring the body’s temperature homeostasis (Figure 1 and Figure 2): the skin temperatures of the ear and tail declined (a decrease in heat loss from the body surface), oxygen consumption and carbon dioxide excretion increased (a heat production increase), and electrical muscle activity strengthened (shivering thermogenesis increased). During the rapid cooling in hypertensive rats, the change in oxygen consumption and carbon dioxide excretion proceeded via two phases, as reported for normotensive animals [16,18]. The decrease in the respiratory exchange ratio that we observed at the end of the cooling meant increased utilization of lipids as an energy source.

The pharmacological activation of the TRPM8 ion channel by menthol in the rats did not alter the latency periods of thermoregulatory reactions either during the rapid or during the slow cooling (Figure 1A and Figure 2A, respectively). This pharmacological activation also did not affect the threshold temperatures of thermoregulatory reactions (Figure 1B,C; Figure 2B,C). Furthermore, the maximum changes in the parameters characterizing heat production and heat loss in these animals also did not change after the treatment with menthol compared to the treatment with saline (Figure 1D–F; Figure 2D–F).

### 2.2. mRNA Expression of Thermosensitive TRP Channels

The experimental effects on the rat organism did not influence the mRNA level of the housekeeping gene under study: peptidyl-prolyl cis-trans isomerase A (*Ppia*) (F_5.105_ < 1) (Appendix A). This finding allowed us to use the expression of this gene as an internal standard when comparing the data between different experimental groups.

#### Thermoneutral Conditions and Application of Saline

The expression levels of the genes of thermosensitive TRP ion channels (*Trpa1*, *Trpm8*, *Trpv1*, *Trpv2*, *Trpv3*, and *Trpv4*) studied in this work are presented in Table 2 and Figure 3. The expression ratios of these genes under the thermoneutral conditions after the application of saline (i.e., in the control animals) matched our earlier results [9]: The *Trpv2* mRNA level was at least an order of magnitude higher than that of the other genes of thermosensitive ion channels, and mRNA levels of *Trpv1* and *Trpv4* were significantly lower in the anterior part of the hypothalamus than in its posterior part (t = 2.24, *p* < 0.05; t = 3.07, *p* < 0.01, for *Trpv1* and *Trpv4*, respectively).

The effects of the menthol-induced activation of the TRPM8 peripheral ion channel. The treatment with menthol under the thermoneutral conditions raised *Trpm8* expression in the anterior hypothalamus of hypertensive rats approximately twice (Figure 3) without any changes in expression of genes of another TRP ion channels. (Table 2). In the posterior part of the hypothalamus, no changes were found in the mRNA expression of the thermosensitive TRP ion channels (Table 2).

The effects of the cold-induced activation of the TRPM8 peripheral ion channel. The rapid cooling of control rats did not affect mRNA expressions of the thermosensitive TRP ion channels either in the anterior or in the posterior parts of the hypothalamus (Table 2, Figure 3). The slow cooling more than doubled *Trpm8* expression in the anterior hypothalamus (Figure 3). The mRNA expression levels of the other genes of thermosensitive TRP ion channels (*Trpv1*, *Trpv2*, *Trpv3*, *Trpv4*, and *Trpa1*) in the anterior hypothalamus did not change in this context (Table 2). In the posterior part of the hypothalamus during slow cooling, there were no alterations in the mRNA expression of the thermosensitive ion channels (Table 2).

The effects of cold exposure after the menthol-induced preactivation of the TRPM8 peripheral ion channel. After the menthol treatment, slow cooling did not cause additional changes in the expression of TRP ion channel genes in the anterior hypothalamus of the rats (Table 2, Figure 3), and the expression of the *Trpm8* gene remained elevated, albeit at the same level as after application of menthol under the thermoneutral conditions. As for rapid cooling after menthol pretreatment, the *Trpm8* mRNA level in the anterior hypothalamus was not different from its level in thermoneutral conditions. (Figure 3). After the preapplication of menthol to the rats, neither the rapid nor slow cooling-induced alterations in the mRNA expression of the thermosensitive TRP ion channels in the posterior part of the hypothalamus (Table 2). 

## 3. Discussion

In this study on hypertensive rats, cooling—both fast and slow—caused a set of reactions aimed to maintain the temperature homeostasis of the body (only hypertensive rats were studied here). These physiological responses developed in the same sequence as that previously described for normotensive animals [16,18]. Our preactivation of the TRPM8 peripheral ion channel with menthol in hypertensive rats did not affect their thermoregulatory responses in the present study. This is their significant difference from normotensive animals, which have been characterized elsewhere [16,17]. It is shown that, in normotensive animals, the application of menthol to the skin, even under thermoneutral conditions, raises oxygen consumption and diminishes the respiratory exchange ratio, whereas upon cooling, the menthol application shortens the latency periods of thermoregulatory reactions and lowers the temperature thresholds at which the reactions are triggered [16,17]. While in hypertensive rats after the activation of peripheral cold-sensitive ion channel TRPM8 with menthol, there were no changes either in the latency periods of thermoregulatory responses or in the threshold temperatures.

Thus, in hypertensive rats (as opposed to normotensive ones), pharmacological activation of the peripheral TRPM8 with menthol does not affect the thermoregulatory responses of these animals. The absence of changes in the physiological reactions to the stimulation of this ion channel suggests that our results are due either to a decrease in the sensitivity of this ion channel or to a reduction in its amount at the site of the stimulus administration. The non-direct confirmation of this may be the previously reported lower expression of *Trpm8* both in the brain (hypothalamus) [9] and at the periphery (in the spleen) [11], which may imply a decrease in the number of TRPM8 ion channels in hypertensive rats.

It should be noted that when investigating the effects of peripheral TRPM8 activation (by the cold or menthol) on the concentration of proinflammatory cytokines in the blood in hypertensive and normotensive animals, we also documented a distinctive response in the normotensive animals—an increase in blood levels of IL-6 and IL-1β—but the absence of such changes in hypertensive animals [19]. These data are consistent with our current conclusion about the smaller amount of the TRPM8 ion channel in hypertensive rats.

In the study of the impact of peripheral ion channel TRPM8 activation by menthol on some parameters of the cardiovascular system [20], a pronounced influence of this channel’s stimulation was detected that was aimed at increasing blood pressure in normotensive animals; in hypertensive animals, these responses were weaker. On the basis of those results and the data about decreased *Trpm8* gene expression in hypertensive animals, the authors concluded that the downregulation of the TRPM8 ion channel may be a compensatory response aimed at preventing a blood pressure surge.

It is known that arterial hypertension is quite often accompanied by changes in lipid metabolism [21,22,23,24,25]. As a rule, these disorders of lipid metabolism include elevated plasma cholesterol concentration and elevation of so-called atherogenic index, i.e., the ratio of low-density to high-density lipoprotein concentrations. It is also reported that the TRPM8 ion channel is related to whole-body fat metabolism. The activation of lipid metabolism by cold exposure has long been beyond doubt [26,27]. Furthermore, that TRPM8 is involved in these processes is confirmed not only by the fact that its thermal activation takes place during cooling but also by data on changes in lipid metabolism under the action of menthol, a specific agonist of this ion channel [16,28,29]. Deficiency of TRPM8 induces numerous metabolic disorders, indicating an alteration of lipid metabolism. For example, *Trpm8* knockout mice [30] have an increased respiratory exchange ratio, increased serum cholesterol levels, and, with aging, these animals become obese. Research on the blood lipid profile in rats with hereditary stress-induced arterial hypertension (ISIAH strain) has revealed lipid metabolism aberrations similar to those in patients with arterial hypertension: a lower level (compared with normotensive animals) of high-density lipoproteins and a higher atherogenic index [31]. It seems likely that these abnormalities of lipid metabolism in hypertensive rats are due to TRPM8 deficiency. Therefore, there is still an open question of what comes first: hypertension and the TRPM8 ion channel downregulation as a compensatory reaction or the deficiency of the TRPM8 ion channel leading to alterations of lipid metabolism and increase in arterial pressure, i.e., to the state of hypertension.

It is noteworthy that the activation of the TRPM8 peripheral ion channel results in a decrease in the atherogenic index. For example, in normotensive animals, a 20 min application of 1% menthol (L-menthol) reduces this parameter from 0.65 ± 0.067 to 0.39 ± 0.051 (*p* = 0.016) (Kozyreva et al., unpublished data). Additionally, it has been shown that in hypertensive animals 5 days after single cold exposure (slow cooling), in other words, after temperature activation of TRPM8, abnormal metabolic indicators (a low level of high-density lipoproteins and a high atherogenic index) return to the level of normotensive animals [31]. Unfortunately, it is not clear how long this beneficial effect lasts, and this question requires additional experiments.

Our study of the influence of peripheral TRPM8 activation on the expression of thermosensitive-ion-channel genes in the hypothalamus uncovered an increase in *Trpm8* gene expression in the anterior hypothalamus after both (1) the temperature action (slow cooling of the rat) and (2) pharmacological treatment (application of menthol to skin). This fact seems to deserve the most attention, because in the anterior hypothalamus of the hypertensive rats, *Trpm8* underexpression is corrected by these treatments to the level of normotensive animals. In the other words, the analyzed treatments normalized one of the characteristics that distinguish hypertensive rats from normotensive ones.

In the text above, we tried to correlate the altered blood lipid profile in hypertensive animals with their underexpression of the TRPM8 ion channel. Using the results of this work and literature data, we can find another correlation: cooling (temperature activation of the TRPM8 ion channel) normalizes both *Trpm8* gene expression in the anterior hypothalamus (measurements within half an hour) and the blood lipid profile (measurements 5 days after the cold exposure) [31]. These data allow us to hypothesize a cause-and-effect relation between the latter two phenomena.

Thus, in this work, it was revealed that, (1) in hypertensive rats, the pharmacological activation of the peripheral TRPM8 ion channel does not affect the analyzed features of maintaining the temperature balance either under thermoneutral conditions or under cold exposure; (2) the expression of the *Trpm8* gene (initially low in the anterior hypothalamus of the hypertensive animals) rises approximately twofold (to the level of normotensive animals) under the influence of (a) slow cooling of the body and (b) pharmacological activation of the peripheral TRPM8 ion channel. The latter fact appears to be the quite important because it enables us to propose a tool for correcting at least some parameters that distinguish a hypertensive state from the normotensive one.

## 4. Materials and Methods

### 4.1. Animals

Males of ISIAH (inherited stress-induced arterial hypertension) rats (*n* = 60), which have high blood pressure [32,33], were used throughout the study. The animals were obtained from the Institute of Cytology and Genetics (the Siberian Branch of the Russian Academy of Sciences, Novosibirsk, Russia). The rats weighed 313.0 ± 4.2 g (mean ± SEM). Their blood pressure was 176.8 ± 0.8 mm Hg. They were housed in groups of five per cage (590 × 380 × 200 mm) at ambient temperatures of 22–24 °C, on a 12 h dark/light cycle, with free access to feed and water. Three-four days before the experiment, the rats were isolated by placement in separate cages to eliminate group effect. All experimental procedures were in compliance with the Directive 2010/63/EU of the European Parliament and were approved by the ethics committee of the Scientific-Research Institute of Neurosciences and Medicine (Novosibirsk, Russia). The applications of solutions, cooling, and registration of physiological parameters were performed on anesthetized animals (nembutal 50 mg/kg) to exclude the effects of movements and of emotional stress.

### 4.2. Cold Exposure

The experiments were conducted at ambient temperatures of 22–24 °C. Cooling was implemented by means of a thermode with an area of 25 cm^2^, applied to the abdominal skin where hair was shaved off beforehand. The removal of hair was carried out by scissors without any chemical drugs and painful irritation. Models of rapid and slow cooling were used. These models differ as follows: There is dynamic activity of cutaneous thermoreceptors during the rapid cooling, but there is no such activity during slow cooling [34,35], and the order of thermoregulatory responses is somewhat different [16,18]. The rate of the skin temperature decrease during the rapid cooling was 0.13 ± 0.01 °C/s, whereas during slow cooling, it was 0.0083 ± 0.0004 °C/s. In all cases, the depth of cooling was the same: a decrease in rectal temperature by 3 °C. 

### 4.3. Nonthermal Modulation of the Afferent Temperature Signal

For this purpose, 1 mL of a 1% suspension of menthol {(-) menthol 5-methyl-2-1- [methylethyl] cyclohexanol; cat. # M-2258, Sigma, Kawasaki, Japan} in saline was applied to the abdominal skin for 20 min. Control animals were subjected to a 20 min application of 1 mL of saline. The solutions for application and the tip of pipette for taking 1 mL were stored at 37 °C to avoid the accidental cooling of the abdominal skin.

### 4.4. Registered Physiological Parameters

In the rats, the following parameters were measured: oxygen consumption and carbon dioxide production (to determine the metabolic rate and to calculate the respiratory exchange ratio, whose changes were used to determine an increase in carbohydrate or lipid metabolism), electrical activity of neck muscles (to assess shivering thermogenesis), rectal temperature (to control core body temperature and to determine the threshold rectal temperature of thermoregulatory reactions), the temperature of the ear and tail skin (to assess the vascular response), and intradermal temperature in the cooled area (to control the rate of cooling and determine threshold skin temperatures of thermoregulatory reactions). All the studied parameters were registered continuously with an MP100 system (set of instruments) (BIOPAC Systems Inc., Goleta, CA, USA). Data preprocessing was performed in the AcqKnowlege software supplied with this system. The following changes during the cooling were defined as threshold values of changes: 0.1 °C for temperature, 1 mL/(min × kg) for oxygen consumption and carbon dioxide excretion, and 1 µV for electrical muscle activity.

### 4.5. Design of the Experiment

Each anesthetized animal was restrained on a thermostated table and subjected to the following manipulations: application of one of the solutions (20 min), measurement of physiological parameters after the application of a solution (“initial state before cooling”) (5 min), cooling of the animal (16.1 ± 0.53 min in the rapid cooling model, 30.4 ± 0.90 min in the slow cooling model), and decapitation. The cooled animals were decapitated when their rectal temperature decline by 3 °C. Control animals (not exposed to the cold) were decapitated when the cooling was initiated in the experimental group.

### 4.6. Experimental and Control Groups of Animals

Six experimental groups of rats were set up: groups 1 and 2, animals treated with either saline or the 1% suspension of menthol; groups 3 and 4, animals subjected to rapid cooling after the application of either saline or menthol; groups 5 and 6, animals subjected to slow cooling after the application of either saline or menthol. There were 10 animals in each group, and only hypertensive rats were studied.

### 4.7. Collection of the Samples for PCR Analysis

After decapitation, rat brains were rapidly removed on ice and the hypothalamic area, including the *tuber cinereum* with its preoptic part, and mamillary bodies were excised at approximately 1.5 mm depth (the *chiasma opticus* and *hypophysis* were removed previously). The hypothalamus was subdivided into anterior and posterior parts by vertical cutting in the middle of the sample. Then, the tissue samples were placed in sterile tubes, frozen under liquid nitrogen, and stored at −70 °C until total RNA extraction.

### 4.8. Quantitative RT-PCR

The expression of the TRP ion channel genes was assayed by the method of quantitative RT-PCR, described in detail previously [9,36,37,38]. The absence of contamination with genomic DNA in the cDNA samples was checked by PCR with primers for tryptophan-hydroxylase I (*Tph1*; this gene is not expressed in the brain) at the maximal number of cycles used in this study. All the primers utilized in this study were composed on the base of the sequences published in the European Molecular Biology Laboratory (EMBL) nucleotide database and were synthesized in the Biosan Company (Novosibirsk, Russia). The sequences of the primers, their characteristics, and the concentrations of genomic DNA used as an external standard for the quantitative determination of the mRNA level of thermosensitive TRP ion channels’ genes in the rat hypothalamus have been published earlier [9,36,37,38]. The expression levels of the TRP ion channel genes are presented as a number of TRP ion channel gene mRNA copies per 100 copies of mRNA of a housekeeping gene (peptidyl-prolyl cis-trans isomerase A; *Ppia*), i.e., in copies/100 *Ppia* copies. The genomic *Tph1* fragment was undetectable in the samples used in the present study.

### 4.9. Statistics

Statistica 8 software package (StatSoft Russia) and Microsoft Excel software were employed for statistical analysis. The data are presented as mean ± SEM and were subjected to Student’s *t*-test (for comparison of identical parameters in control (saline application) and experimental (menthol suspension application) groups under the same temperature conditions; for comparison of identical parameters in saline application (or menthol suspension application) groups under different temperature conditions) and to multivariate analysis of variance (ANOVA) followed by Fisher’s post hoc test for multigroup comparisons (the comparison of house-keeping gene expression in the different parts of hypothalamus in the animals of different experimental groups). A difference was considered significant at *p* < 0.05.

## 5. Conclusions

The results obtained in this study, together with the data provided during their discussion, indicate the important value of TRPM8 both in the elevation of blood pressure and in metabolic changes that can provoke hypertension as a metabolic disease. The activation of peripheral TRPM8 (by menthol or cooling) contributes to the normalization of some metabolic processes as well as the expression of the gene of this ion channel in hypothalamus. The activation of TRPM8 located in the gastrointestinal tract [39,40] leads to the diminution of blood pressure. Therefore, the activation of TRPM8 with different localization may variously change physiological parameters. Undoubtedly, comprehensive research studies of TRPM8 open up new opportunities in findings of optimal natural impacts on the organism (by temperature or by natural plant component) to combat hypertension.

## Figures and Tables

**Figure 1 ijms-23-06088-f001:**
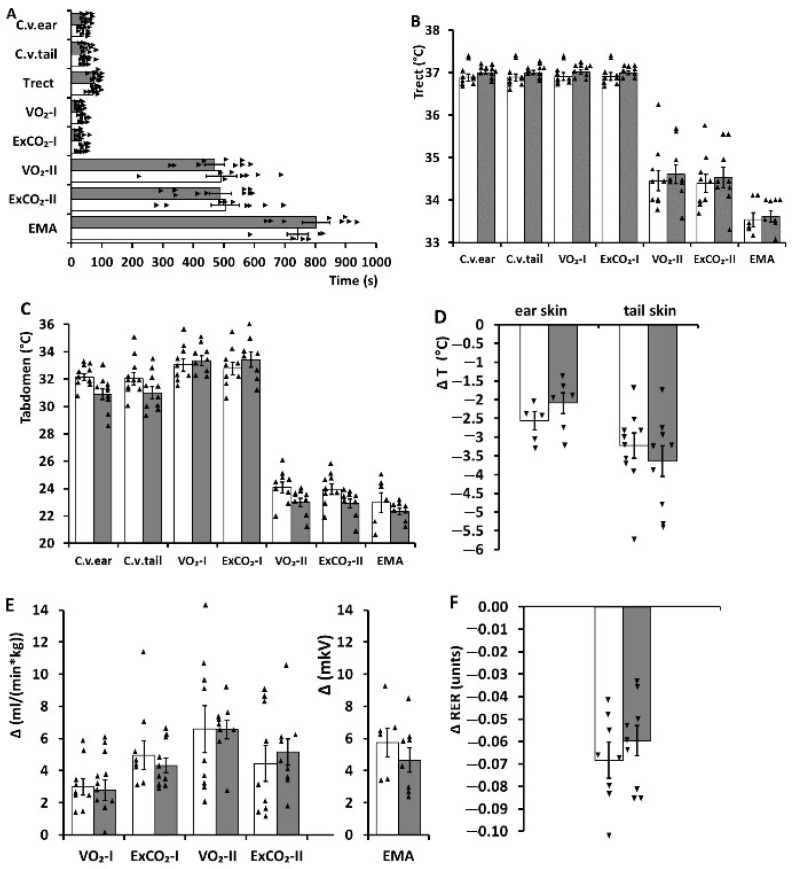
Parameters of thermoregulatory responses to rapid cooling in hypertensive ISIAH rats subjected to the peripheral ion channel TRPM8 activation by menthol (application of a 1% suspension in saline to the skin) and control hypertensive ISIAH rats (application of saline). (**A**) Latency periods (s) of thermoregulatory reactions during the rapid cooling. Legend: C.v.ear, the constrictor reaction of ear skin vessels; C.v.tail, the constrictor reaction of tail skin vessels; T rect, the decrease in rectal temperature; VO_2_-I, an oxygen consumption increase in the first phase of the metabolic response; ExCO_2_-I, an increase in carbon dioxide excretion in the first phase of the metabolic response; VO_2_-II, an oxygen consumption increase in the second phase of the metabolic response; ExCO_2_-II, an increase in carbon dioxide excretion in the second phase of the metabolic response; EMA, an increase in muscle electrical activity. (**B**) Threshold values of core body temperature (rectal; °C) for the initiations of thermoregulatory responses during rapid cooling. Legend: the same as in (**A**). (**C**) Threshold values of the abdominal skin temperature (°C) for the initiations of thermoregulatory responses during rapid cooling. Legend: the same as in (**A**). (**D**) Maximum changes in heat loss parameters (changes in the ear and tail skin temperatures) during rapid cooling. (**E**) Maximum changes in the parameters of heat production during rapid cooling. Legend: the same as in (**A**). (**F**) Maximum changes in the respiratory exchange ratio during rapid cooling. Triangles represent the individual values of the parameters. In all panels, the data from control animals are represented by open bars, and data from the animals subjected to the peripheral ion channel TRPM8 activation with menthol are indicated by filled bars. Thermoregulatory responses were compared by the *t*-test (Statistica 8 software package). *p*-values are presented in Appendix A.

**Figure 2 ijms-23-06088-f002:**
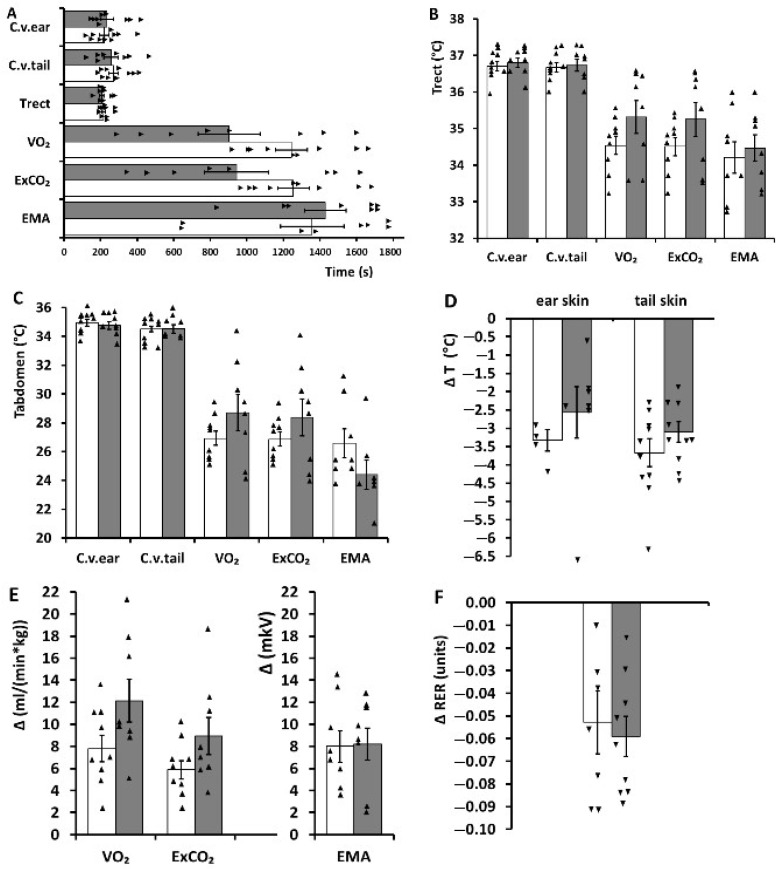
Parameters of thermoregulatory responses to slow cooling in hypertensive ISIAH rats subjected to the peripheral ion channel TRPM8 activation by menthol (application of a 1% suspension in saline to the skin) and control hypertensive ISIAH rats (application of saline). (**A**) Latency periods (s) of thermoregulatory reactions during the slow cooling. Legend: C.v.ear, the constrictor reaction of ear skin vessels; C.v.tail, the constrictor reaction of tail skin vessels; T rect, the decrease in rectal temperature; VO_2,_ an oxygen consumption increase; ExCO_2,_ an increase in carbon dioxide excretion; EMA, an increase in muscle electrical activity. (**B**) Threshold values of core body temperature (rectal; °C) for the initiations of thermoregulatory reactions during slow cooling. Legend: the same as in (**A**). (**C**) Threshold values of the abdominal skin temperature (°C) for the initiations of thermoregulatory responses during slow cooling. Legend: the same as in in (**A**). (**D**) Maximum changes in heat loss parameters (changes in the ear and tail skin temperatures) during slow cooling. (**E**) Maximum changes in the parameters of heat production during slow cooling. Legend: the same as in in (**A**). (**F**) Maximum changes in the respiratory exchange ratio during slow cooling. Triangles represent the individual values of the parameters. In all panels, the data from control animals are represented by open bars, and data from the animals subjected to the peripheral ion channel TRPM8 activation with menthol are indicated by filled bars. Thermoregulatory responses were compared by the *t*-test (Statistica 8 software package). *p*-values are presented in Appendix A.

**Figure 3 ijms-23-06088-f003:**
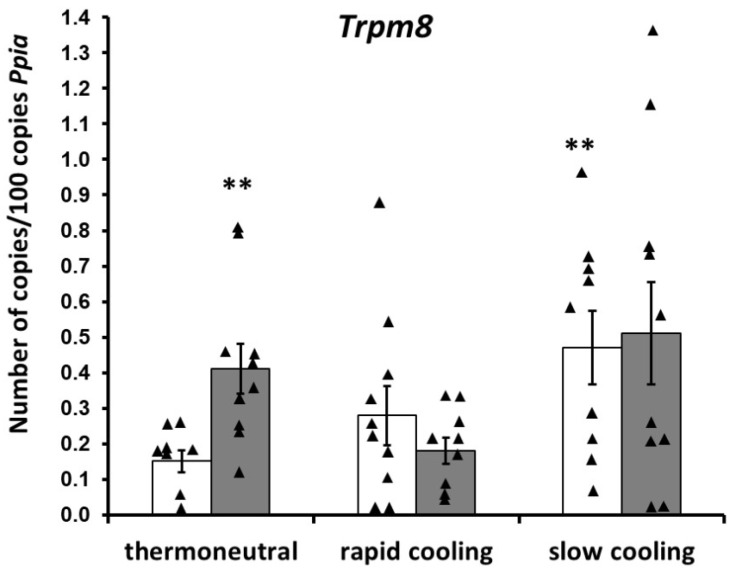
Effect of the peripheral ion channel TRPM8 activation on mRNA expression of the *Trpm8* gene in the anterior hypothalamus of hypertensive rats. Pharmacological activation of the TRPM8 ion channel: application of a 1% menthol suspension. Thermal activation of the TRPM8 ion channel: cooling (rapid or slow). ** *p* < 0.02: a significant difference from the control (application of saline) under thermoneutral conditions (under thermoneutral conditions for comparison saline-menthol *p* = 0.007, t = 3.08; between control (saline) under thermoneutral conditions and saline after slow cooling *p* = 0.013, t = 2.79; between control (saline) under thermoneutral conditions and saline after rapid cooling *p* = 0.207, t = 1.32; between saline and menthol after rapid cooling *p* = 0.305, t = 1.06; between saline and menthol after slow cooling *p* = 0.831, t = 0.216).Triangles represent the individual values. Open bars—application of saline; filled bars—application of a 1% menthol suspension.

**Table 1 ijms-23-06088-t001:** Comparison of the temperature homeostasis parameters under thermoneutral conditions in hypertensive rats after the application of either saline (control, *n* = 30) or a 1% menthol suspension in saline (*n* = 30). Parameters were compared by the *t*-test (Statistica 8 software package).

Temperature Homeostasis Parameters	Control	Menthol Application	*p* Value
Ear skin temperature, °C	28.5 ± 0.14	28.6 ± 0.23	>0.05 (0.566)
Tail skin temperature, °C	30.3 ± 0.14	30.1 ± 0.15	>0.05 (0.177)
Abdominal skin temperature, °C	36.8 ± 0.04	36.8 ± 0.04	>0.05 (0.847)
Rectal temperature, °C	36.9 ± 0.06	37.0 ± 0.06	>0.05 (0.478)
Oxygen consumption, mL/(min × kg)	18.6 ± 0.32	18.9 ± 0.35	>0.05 (0.456)
Carbon dioxide excretion, mL/(min × kg)	15.5 ± 0.43	15.4 ± 0.57	>0.05 (0.849)
Respiratory exchange ratio, dimensionless units	0.84 ± 0.019	0.81 ± 0.023	>0.05 (0.365)
Muscle electrical activity, µV	1.2 ± 0.06	1.4 ± 0.09	>0.05 (0.051)

**Table 2 ijms-23-06088-t002:** mRNA levels of the genes of thermosensitive TRP ion channels in the anterior and posterior parts of the hypothalamus in hypertensive rats, control animals and animals subjected to the peripheral TRPM8 ion channel activation by menthol under thermoneutral conditions and with subsequent cooling. Number of rats is 10 for each group.

Treatment Group	mRNA Levels of Thermosensitive TRP Ion Channel Genes (Number of Copies per 100 Copies of *Ppia* mRNA)										
	Anterior Hypothalamus					Posterior Hypothalamus					
	*Trpa1*	*Trpv1*	*Trpv2*	*Trpv3*	*Trpv4*	*Trpa1*	*Trpm8*	*Trpv1*	*Trpv2*	*Trpv3*	*Trpv4*
**Thermoneutral Conditions**											
**Control**	1.5 ± 0.53	2.0 ± 0.65	97.7 ± 11.08	1.8 ± 0.44	0.11 ± 0.05	2.2 ± 0.43	0.2 ± 0.07	4.9 ± 1.05	98.1 ± 12.58	1.5 ± 0.18	0.3 ± 0.05
**Menthol application**	1.3 ± 0.29	1.8 ± 0.38	92.3 ± 8.92	1.8 ± 0.27	0.12 ± 0.04	2.4 ± 0.19	0.2 ± 0.05	6.8 ± 1.59	95.4 ± 11.05	1.4 ± 0.22	0.3 ± 0.04
***p* value**	0.701	0.752	0.706	0.999	0.793	0.667	0.764	0.343	0.872	0.809	0.779
**Rapid Cooling**											
**Control**	1.2 ± 0.26	1.8 ± 0.55	83.5 ± 7.67	1.8 ± 0.21	0.11 ± 0.04	2.6 ± 0.26	0.1 ± 0.02	6.0 ± 1.80	86.8 ± 14.30	1.6 ± 0.15	0.2 ± 0.03
**Menthol application**	1.9 ± 0.42	2.0 ± 0.38	85.5 ± 7.71	1.7 ± 0.19	0.2 ± 0.04	2.7 ± 0.33	0.2 ± 0.06	4.2 ± 0.69	100.2 ± 11.10	2.0 ± 0.29	0.3 ± 0.05
***p* value**	0.156	0.766	0.855	0.780	0.147	0.859	0.148	0.344	0.464	0.314	0.401
**Slow Cooling**											
**Control**	1.3 ± 0.28	1.9 ± 0.33	76.3 ± 9.96	1.5 ± 0.32	0.2 ± 0.06	2.5 ± 0.29	0.2 ± 0.06	4.6 ± 1.28	80.6 ± 9.29	1.7 ± 0.13	0.3 ± 0.05
**Menthol application**	1.1 ± 0.25	2.1 ± 0.36	85.6 ± 7.22	2.4 ± 0.39	0.3 ± 0.09	2.3 ± 0.42	0.2 ± 0.04	5.3 ± 0.94	87.3 ± 9.93	1.8 ± 0.19	0.3 ± 0.05
***p* value**	0.539	0.712	0.456	0.096	0.414	0.799	0.576	0.691	0.632	0.624	0.961

## Data Availability

All data are in the work logs (Department’s Archive).

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
