# Peer review of "Effect of Skin Ion Channel TRPM8 Activation by Cold and Menthol on Thermoregulation and the Expression of Genes of Thermosensitive TRP Ion Channels in the Hypothalamus of Hypertensive Rats"

_ijms, 2022, doi:10.3390/ijms23116088_

Round 1

Reviewer 1 Report

The submitted manuscript is a very interesting example of the article taking into consideration the activity of thermoregulated TRP ion channels. Introduction part including a description of TRP is well prepared.

Material and methods are well and adequately described. 

The Results are properly described. Is that possible to calculate and add on the figure 1 and 2 the information about p-value? Please add in the description of the figures the name of the used statistical test. Please add these information for Table 2 as well. Please specify in the paper (section 4.9 or description of the appropriate table or figure) for which statistical analyses the Authors used t-test and ANOVA.

The Discussion section is prepared in a thoughtful way and a sufficient number of articles was cited.

Summarizing, I recommend this manuscript for publication when the Authors correct the abovementioned issues in the article.

Author Response

We thank the Reviewer for his sincere interest to our article and useful comments.

According to remark of the distinguished reviewer, we :

1.have prepared the table (Table S1) for P-value for comparisons of all physiological parameters presented in Figure 1 and Figure 2;

2.have completed the Table 2 by P-value for comparisons of gene expression in different condition;

3.have indicated (in section 4.9) the statistical tests used for comparisons of the studied parameters

Reviewer 2 Report

The thermosensitive ion channel TRPM8 has attracted many attentions since it was discovered. The authors presented their studies on the TRPM8 activation and the expression of genes of thermosensitive TRP ion channels in the hypothalamus of hypertensive rats. The results are interesting, while there are some questions should be clarified:

  1. There are 3 formats for TRPM8 in the manuscript, “TRPM8”, “Trpm8” and “Trpm8”, what’s the difference?
  2. Page 2, line 66, a closed bracket is missed. Page 3, line 83, “Figure 1, 2” should be “Figures 1, 2”. Page 8, line 180, there is one dot on each side of (Figure 3).
  3. 1, what do the triangles mean? There is no induction on this. In the Fig. 1 (D) the ear skin and the tail skin show opposite results, how to understand? In Page 6, lines129-131, the authors claim that “the maximum changes in the parameters…. did not change after the treatment with menthol compared to the treatment with saline”, but it looks there is some change?
  4. There are 36 references, however, only 6 of them are published after 2017. There should be more references from recent 5 years.
  5. Table 2, why there is no TRPM8 in the anterior hypothalamus? For the TRPV2, it has more significant changes for Thermoneutral and Slow cooling conditions compared to other ion channels, while for rapid cooling, the change of TRPV2 is less significant compared to other ion channels. How to understand this?

Author Response

We thank the Reviewer for his sincere interest to our article and useful comments.

  1. According to “Guidelines for Formatting Gene and Protein Names” (https://www.biosciencewriters.com/Guidelines-for-Formatting-Gene-and-Protein-Names.aspx) for mice and rats, “Gene symbols are italicized, with only the first letter in upper-case (e.g., Gfap). Protein symbols are not italicized, and all letters are in upper-case (e.g., GFAP)”. So, in our article Trpm8 is a name of gene, TRPM8 is a name of protein (ion channel). “Trpm8” is mistake (thank you for indication it). We have corrected it.
  2. We have corrected all the mistakes indicated by the distinguished reviewer.
  3. The triangles in the Figures represent the individual values of the parameters. We have added this explication to the legends of Figures.

2.The panel D (in Figures 1 and 2) represents maximum changes in heat loss parameters (changes in the ear and tail skin  temperatures) during cooling (rapid or slow). Both ear and tail skin temperatures dropped under these conditions. Difference in solutions applied (saline or menthol suspension) did not alter the direction of temperature changes. The exact values for decrease of skin temperature in one of panels, were the distinguished reviewer had seen the “opposite results”, are:

for ear skin temperature -2.56±0.24ºC after saline application and -2.09±0.28ºC after menthol suspension application (P>0.05);

for tail skin temperature -3.22±0.34ºC after saline application and -3.64±0.41ºC after menthol suspension application (P>0.05).

There is no statistical difference between saline and menthol suspension applications both for ear skin temperature or tail skin temperature. Thus, the results are the same.

The exact values for decrease of skin temperature in the other D panels are:

for ear skin temperature -3.33±0.29ºC after saline application and -2.57±0.70ºC after menthol suspension application (P>0.05);

for tail skin temperature -3.67±0.38ºC after saline application and -3.10±0.28ºC after menthol suspension application (P>0.05).

In this panel there is no statistical difference between saline and menthol suspension applications both for ear skin temperature or tail skin temperature too.

  1. According to remark of the distinguished reviewer we have added 3 references after 2017. Unfortunately, there is no reference after 2017 on the subject we studied.
  2. Yes, there is no TRPM8 in the anterior hypothalamus in Table 2 because these data are presented in Figure 3.

There is no statistical difference between the values of TRPV2 ion channel gene expression in different conditions.

P-values for comparison of TRPV2 ion channel gene expression in anterior and posterior parts of hypothalamus under different experimental conditions are presented below.

in anterior hypothalamus:

Application of saline: Thermoneutral condition vs Rapid cooling  P= 0.295651

Application of saline: Thermoneutral condition vs Slow cooling  P= 0.171505

Rapid cooling: application of saline vs application of menthol suspension  P= 0.854515

Slow cooling: application of saline vs application of menthol suspension  P= 0.456103

in posterior hypothalamus:

Application of saline: Thermoneutral condition vs Rapid cooling  P= 0.558709

Application of saline: Thermoneutral condition vs Slow cooling  P= 0.278740

Rapid cooling: application of saline vs application of menthol suspension  P= 0.463957

Slow cooling: application of saline vs application of menthol suspension  P= 0.632214

Reviewer 3 Report

In this work, the authors studied the effects of temperature and menthol activation of the TRPM8 receptors expressed in the skin of hypertensive rats on physiological parameters related to temperature control and the expression of mRNAs of different types of TRP in the hypothalamus. The results are interesting, but the lack of controls makes their interpretation difficult.

Main Issues:

1.- In materials and methods, the authors established that they used inherited stress-induced arterial hypertension in male rats, which has increased blood pressure, and they showed a couple of references to support this statement. However, it is not clear how the authors developed this model. Then, despite giving references, I think it is very necessary to show that these animals present high systolic and diastolic pressures to empirically confirm that they are actually hypertensive.

2.- Then is the stress used to generate this hypertensive model the cause of the decrease of the TRPM8 mRNA in the hypothalamus or is the hypertension by itself? If so, do animals with a high NaCl diet show a similar result?

3.- These animals have comparable levels or activity of the TRPM8 in the skin comparable to the normotensive animals? I think western blot or inmmunofluoresce studies are necessary.

4.- A graphical model of how the activation of skin sensory neurons (which supposedly express the TRPM8) modulates the expression of the same receptors in neurons and/or astrocytes of the hypothalamus would be very useful.

5.-How can the expression of TRPM8 receptors in the hypothalamus be involved in temperature control ? I understand that the brain is one of the organs that best maintains temperature homeostasis; therefore, what is the sign that triggers its opening in the cells of the central nervous system?

6.- A western blot or inmmunofluorescence study would add much information about the expression of TRPM8 in this brain area... sometimes there is no correlation between the mRNA level with its protein level.

Minor

1.- In line 89, the authors declare that "The decrease in the respiratory exchange ratio that we observed at the end of the cooling meant increased use of lipids as an energy source." Please give a reference that supports this idea.

2.- The phrase of lines 106 and 107, please leave it in the legend of the figure.

3.- The phrase of lines 122 and 123, please leave it in the legend of the figure.

Author Response

We thank the Reviewer for his sincere interest to our article and useful comments.

The distinguished reviewer in the description of the work noted: “… the lack of controls makes the results’ interpretation difficult”.

We can’t agree with this statement.

In this work, in hypertensive animals, we studied physiological reactions (including gene expression) in response to the activation, thermal or pharmacological, of the thermosensitive ion channel TRPM8. The groups of hypertensive animals without peripheral TRPM8 activation were the control ones. Experiments with the same design on normotensive animals were performed earlier (ref #12, 13, 17) and served as background for present investigation.

Main issues:

  1. Used in our study ISIAH rats are characterized by high blood pressure. The name of the strain “Inherited Stress-Induced Arterial Hypertension” indicates, that the main factor for selection the animals at the creation of this strain was the increasing of blood pressure in response to stress (restriction of conscious rats for 30 min). The word “Inherited” is also very important. It indicates that at the process of strain creation, the high blood pressure became the internal characteristic of rats that is inherited from parents to descendants.

Currently, in rats of this line, the mean basal (at a rest) blood pressure is 160-170 mm Hg. in males and 145-150 mm Hg. in females. Systolic blood pressure during stress increases compared to the basal level and averages 200 and 175 mm Hg. in males and females,  respectively (Markel, 1992).

Male rats in our study were actually hypertensive. The level of blood pressure indicated in section 4.1 (176.8±0.8 mm Hg) is mean ± SEM of the values of blood pressure empirically measured immediately before the handing rats for our experiments.

As for us, we did not stress the animals in aim to raise their blood pressure. Moreover, we did everything to avoid stress: experiments were performed on anesthetized animals. The high blood pressure is internal characteristic of the ISIAH rats and it was enough for us.

  1. In section 3, we discuss the question of what comes first: hypertension and the TRPM8 ion channel downregulation as a compensatory reaction or the deficiency of the TRPM8 ion channel leading to alterations of lipid metabolism and increase in arterial pressure i.e. to the state of hypertension. So, if the hypertension comes first, it is possible to expect the TRPM8 ion channel downregulation in different hypertension models and in the animals with a high NaCl diet too.
  2. The question posed by the distinguished reviewer is very interesting and in order to resolve it, western blot or inmmunofluoresce studies are really useful.

But as for our study, the question posed was the follow: does reduced level of Trpm8 mRNA in the anterior hypothalamus find in hypertensive animals affect their physiology, in particular, their thermoregulatory responses and mRNA expression of thermosensitive TRP ion channels in the hypothalamus in response to stimulation (temperature and pharmacological) of peripheral ion channel TRPM8?

So, the aim of our study was to find out the response to this question.

  1. We agree that graphical model demanded by the distinguished reviewer may be useful. But to date there is no enough information to do it.

Some aspects of how the activation of skin sensory neurons modulates the activity of neurons and/or astrocytes in the hypothalamus are presented in many articles of Handbook of Clinical Neurology, 3rd series. Thermoregulation: From Basic Neuroscience to Clinical Neurology, Romanovsky AA (ed) (in graphical format too). After the afferent information reaches the hypothalamus, the formation of an efferent signal begins. This process includes many changes in neurons function. Changes in expression of some genes are among them.

But now we have no exact answer for the question why does the expression of Trpm8 in hypothalamus change in response to the activation of skin TRPM8? Perhaps, it is only one of the genes whicht expression changes in response to activation of skin TRPM8.

  1. The thermal threshold for the activation of TRPM8 ion channel (<28 ºC) is out of the physiological temperature range and, of course, there appears the question about its role in the brain. Evidently, its role in brain is not restricted by the response to temperature stimulus. The latter studies of this ion channel demonstrated its role in some other physiological functions, for example influence on the GABA system (in hippocampus, Zhang et al., 2008) and pain sensation. As for its activation, TRPM8 ion channel may be activated (in addition to activation by temperature and cooling compounds, such as menthol or icilin) by voltage and hyperosmolarity (see Izquierdo et al., 2021). Lipids and enzymes of lipid metabolism can also modulate its activity (Yudin and Rohacs, 2012;Morales-Lázaro et al., 2017). So, the role and peculiarities of function of TRPM8 ion channel in brain and, in particular in hypothalamus, wait their investigators.
  2. We completely agree with the distinguished reviewer that a western blot or inmmunofluorescence study add much information about TRPM8, but in case when the aim question is to study this ion channel, the quantity of this ion channel, ratio of its protein and its gene expression, its peculiarities of function.

In our case, the question was the other (see answer N3).

Minor

1 We added the references (ref #14, 15) after the phrase "The decrease in the respiratory exchange ratio that we observed at the end of the cooling meant increased use of lipids as an energy source."

  1. Yes, the phrase of lines 106 and 107 belongs to the legend of the figure. We corrected this mistake.
  2. Yes, the phrase of lines 122 and 123 belongs to the legend of the figure. We corrected this mistake.

Round 2

Reviewer 3 Report

Dear Authors,

Thank you for sharing your points of view and accept my comments in a good way.

Only one more thing, please add this " The rats weighed 313.0±4.2 g (mean ± SEM). 291 Their blood pressure was 176.8±0.8 mm Hg. " somewhere in section 2.1. 

Author Response

We thank the Reviewer for his additional comments and recommendation.

In accordance with his recommendations we add the description of the animals used (their weight and blood pressure) at the beginning of section Results.

Page 2, section 2, line 73. The sentences: «Hypertensive rats used in our study weighed 313.0±4.2 g (mean ± SEM). Their blood pressure was 176.8±0.8 mm Hg (mean ± SEM).» are added to the main text.